# CNT-Based Solar Thermal Coatings: Absorptance vs. Emittance

**Yelena Vinetsky, Jyothi Jambu, Daniel Mandler \*** and **Shlomo Magdassi \***

Institute of Chemistry, The Hebrew University of Jerusalem, Jerusalem 9190401, Israel; yelenavi@gmail.com (Y.V.); jyothi.jambu@gmail.com (J.J.)
**\*** Correspondence: daniel.mandler@mail.huji.ac.il (D.M.); magdassi@mail.huji.ac.il (S.M.)

**Abstract:** A novel approach for fabricating selective absorbing coatings based on carbon nanotubes (CNTs) for mid-temperature solar–thermal application is presented. The developed formulations are dispersions of CNTs in water or solvents. Being coated on stainless steel (SS) by spraying, these formulations provide good characteristics of solar absorptance. The effect of CNT concentration and the type of the binder and its ratios to the CNT were investigated. Coatings based on water dispersions give higher adsorption, but solvent-based coatings enable achieving lower emittance. Interestingly, the binder was found to be responsible for the high emittance, yet, it is essential for obtaining good adhesion to the SS substrate. The best performance of the coatings requires adjusting the concentration of the CNTs and their ratio to the binder to obtain the highest absorptance with excellent adhesion; high absorptance is obtained at high CNT concentration, while good adhesion requires a minimum ratio between the binder/CNT; however, increasing the binder concentration increases the emissivity. The best coatings have an absorptance of ca. 90% with an emittance of ca. 0.3 and excellent adhesion to stainless steel.

**Keywords:** carbon nanotubes (CNTs); binder; dispersion; solar thermal coating; absorptance; emittance; adhesion; selectivity

## 1. Introduction

Solar energy conversion by concentrated solar power (CSP) systems is a leading technology in renewable energies. According to the Strategic Energy Technology (SET) Plan, the goal is that by 2030, up to 27% of EU electricity will be generated by solar energy. In that report, the global electricity share of CSP is envisioned to reach 11% by 2050 [1].

The spectrally selective solar coatings (SSCs) of the collector are an essential element in increasing the efficiency of mid-range temperature CSP systems for non-evacuated solar-trough or linear Fresnel collectors.

An appealing material for solar absorbers, which is still not fully explored, is carbon nanotubes (CNTs) [2]. CNTs are known for their unique optical, electrical, and mechanical properties, and are considered as one of the best light-absorbing materials [3,4]. Moreover, CNTs exhibit high physical and chemical stability, a large surface area, and high thermal conductivity [5].

However, to obtain very high absorptance, >95%, which is a prerequisite for efficient solar thermal coatings, the CNTs must be grown from the vapor phase, which is suitable only for small-scale devices, by using costly equipment. Wang et al. reported the visible and the near-infrared radiative properties of vertically aligned, multiwalled carbon nanotubes [6]. The results showed that CNTs do not only reflect light weakly but also strongly absorb it. These combined features make CNTs an ideal candidate for realizing a "super black coating." A CNT forest (composed of densely aligned nanotubes grown perpendicularly to the surface) has been used as an optical coating [7,8]. This type of coating shows

good light absorption and anti-reflective performance, but its production requires unique equipment and, specific conditions are required for chemical vapor deposition (CVD), such as high temperature and pressure. This process is expensive and has several drawbacks, including limited type substrates, small collector area, and poor adhesion to the substrate.

Mizuno et al. fabricated vertically aligned CNTs that showed an extremely high absorptance of 0.98–0.99 [9]. Cao et al. constructed a tandem structure of aligned CNTs on Au, which exhibited an absorptance of 0.95 with no spectral selectivity [10]. Selvakumar et al. reported an absorptance of 0.95 and remarkably low emittance of 0.2 using a CNT-based tandem absorber fabricated by CVD [11,12].

The application of carbon nanotubes in solar collectors was reported by several researchers, as will be briefly described here. Abdelkader et al. reported the fabrication of solar selective coating for Flat plate solar air heaters (SAHs) by embedding CNTs and cupric oxide nanoparticle (CuO) into the black paint [13]. The coating showed high solar selective properties with solar absorptance and thermal emittance reaching 0.964 and 0.124, respectively. Li et al. demonstrated a high performance solar thermoelectric generator (STEG) system combined with solar concentrators and a carbon nanotubes absorber, which can greatly improve the solar–thermal conversion process [14]. The enhanced efficiency is ensured by the optimized system thermodynamics as a result of the combination of solar concentration devices and the CNT-based solar absorber. Sobhansarbandi et al. confirmed that CNT sheet coatings improved the solar energy phase change and also increased solar energy absorption [15]. In a different study, it was found that CNTs can increase the evaporation efficiency of a steam generation [16]. Kasaeian et al. evaluated the effect of carbon nanotube/ethylene glycol on a direct absorber solar collector attached to a parabolic trough [17]. The optical efficiency was 71.4%, and the thermal efficiency of the collector was 17% higher than that of the base fluid. Multiwalled carbon nanotubes (MWCNTs) were found to absorb almost 100% of solar energy as compared to other nanoparticles, making them ideal for application in a direct absorber solar collector (DASC) due to their smaller agglomeration and high stability [18]. Shende et al. investigated the DASC using MWCNTs as a working fluid and compared the behavior in water and ethylene glycol [19].

Wet deposition processes with CNTs as a functional material were also reported. For example, Rincon et al. fabricated spectrally selective coatings using a CNT–$TiO_2$ composite by a sol-gel process combined with dip-coating, but these coatings exhibited low absorptance [20]. Roro et al. fabricated multiwalled CNT–nickel oxide nanocomposite coatings also by dip-coating, yielding an absorptance of 0.84 and emittance of 0.2 [2]. From the literature survey, it is evident that most of the CNT-based coatings suffer from low spectral selectivity.

Making coatings composed of CNTs brings a major challenge in dispersing them in liquids. Bibi et al. reported the treatment of multiwalled carbon nanotubes with gamma irradiation in the air or by using dilute acids ($H_2SO_4$/$HNO_3$) combined with 20 kHz ultrasound to compare their dispersion [21,22]. The CNT microstructure was investigated using transmission electron microscopy, which revealed that both methods effectively modified the CNTs to overcome aggregation of the nanotubes, resulting in efficient dispersion in ethanol. Raiah et al. reported that imidazolium-based ionic liquids with a long hydrocarbon chain enable high concentration dispersion of double-walled carbon nanotubes (DWCNTs) in water, having a low viscosity [23]. Dispersibility of DWCNTs increased as the length of the hydrocarbon chain increased. The suspensions of partially de-bundled DWCNTs proved to be stable for over a month. Ferreira et al. studied ionic liquid suspensions of CNTs functionalized with carboxylic and alkane groups in various solvents [24]. CNTs were functionalized using $H_2SO_4$/$HNO_3$ and subsequently functionalized by dodecylamine (DDA). It was found that the dispersion stability was strongly dependent on the solvent and carbon nanotube surface interactions, which can vary with the chemical nature of the solvent. The work of Lee et al. [25] provided a comparative investigation of the thermal efficiency of a volumetric receiver using MWCNT–water nanofluid and a surface receiver with conventional fluid. The study demonstrated that the volumetric solar receivers (VRs) with water-based MWCNTs can achieve higher efficiency, compared to the conventional surface-based solar receivers over 10%.

Previous liquid formulations developed in our group for solar selective coatings [26–28] were directly applied on Inconel or aluminum substrates by spraying. The coatings had a multilayer structure composed of different materials with specific optical properties. Additionally, a self-cleaning CNT-based near-perfect solar absorber coating was fabricated for non-evacuated CSP applications [29]. The coating shows high durability and is super hydrophilic (0° contact angle) after plasma treatment, without affecting the solar absorptance and excellent coating adhesion.

In the previous publication, a multilayer coating was presented, in which the first absorbing layer was made of CNTs, which gave high absorptance but also high emittance. The second layer was an infrared (IR)-reflecting layer composed of indium tin oxide (ITO). The ITO layer enabled the reduction of the emittance; however, it simultaneously reduced the absorptance. The third layer, which functioned as an anti-reflecting layer, was made of boehmite (AlOOH), whose presence made it possible to improve the absorptance without a significant change in the emittance. The outcome was a multilayer CNT/ITO/AlOOH coating with very good spectral characteristics. However, the fabrication of the multilayer coating is complex, and a simpler solution is required.

Here we present novel formulations for a one-step fabrication of a single-layer CNT coating on stainless steel (SS), which is commonly used for the manufacturing of solar collector pipes. The coatings are made by water- or solvent-based liquid formulations composed of dispersed CNTs and a ceramic binder. We studied the effect of CNTs and the binder type and their concentrations on the absorptance, emittance, and adhesion of the thin coatings.

## 2. Materials and Methods

### 2.1. Materials

Nanocyl (NC-7000TM) MWCNTs were purchased from Nanocyl SA, Sambreville, Belgium; Short Cheap Tube MWCNTs (outer diameter 10–20 nm) from Cheap Tubes, Cambridgeport, MA, USA; aluminum oxyhydroxide (AlOOH, boehmite) powder (Disperal P3) from SASOL, Germany; SILRES REN 100 (silicone resin) from Wacker Chemie AG, Munchen, Germany; BYK 333 and BYK 9077 from Byk-Chemie GmbH, Wesel, Germany; Solsperse 46,000 from The Lubrizol Corporation; methyltrimehtoxysilane (MTMS) from Alfa Aesar, Heyshem, UK; and N,N-dimethylformamide (DMF) from Fischer Scientific, Loughborough, UK.

### 2.2. Coating with Water-Based CNTs Dispersion

The first 0.24 g of Cheap Tubes CNTs was placed in a 28 mL vial and 0.12 g of Solsperse 46,000 was added, followed by the addition of 4 g of propylene glycol. Then 15.64 g of distilled water was added, and the mixture was sonicated for 7.5 min at 750 W in pulses for 1 s on and 1 s off with an amplitude of 85% (Sonics Vibra Cell Sonicator, Sonics & Materials, Inc., Newtown, CO, USA).

For binder preparation, 10% of AlOOH was dispersed in distilled water by stirring for 2 h and then 1 g of MTMS was added, followed by stirring for 2.5 h.

For the preparation of the final composition, 5 g of CNT dispersion, 3.65 g of distilled water, 0.1 g of wetting agent (Byk333), and 1.25 g of the binder were mixed and stirred for 24 h. The obtained formulation (2.0 mL) was sprayed onto a heated substrate (60–80 °C) with an area of $5 \times 5$ cm$^2$. The sample was initially heated to 100 °C with a heating rate of 5 °C/min for 30 min. Then the sample was heated to 350 °C with a heating rate of 10 °C/min for 30 min. Finally, it was heated to 400 °C with a heating rate of 10 °C/min for 60 min.

Flat metal substrate was thoroughly washed with aqueous soap and then rinsed with ethanol and acetone.

### 2.3. Coating with Solvent-Based CNT Dispersion

An amount of 0.1 g of Nanocyl CNTs was placed in a 28 mL vial and 2 g of 10% Byk-9077 and 17.9 g of DMF were added. The mixture was sonicated for 20 min at 750 W in pulses of 2 s on and 1 s off with an amplitude of 85% (Sonics Vibra Cell Sonicator, Sonics & Materials, Inc., USA).

An amount of 6 g of Ren 100 (10% in DMF) was added to the CNT dispersion, the obtained mixture was shaken well and then sonicated in a bath for 5 min. The coating was prepared by spraying 5 mL of the mixture onto a heated (100–120 °C) substrate, with an area of $5 \times 5$ cm$^2$. The samples were cured in an oven at 350 °C for 2 h with a heating rate of 15 °C/min.

### 2.4. Absorptance and Emittance Measurements

The reflectance spectra of the coatings were recorded in the region of 280–2500 nm. With a CARY 5000 UV–vis–NIR spectrophotometer (Varian Analytical Instruments, Palo Alto, CA, USA) using an integrated sphere.

The absorptances of the coatings were calculated from the UV–vis–NIR (NIR: near-infrared) reflectance spectra using the equation $\alpha + R + T = 1$, in which $\alpha$ is the absorptance, R is the reflectance, and T is the transmittance. As the coatings were fabricated on a stainless steel substrate, the transmittance is zero and therefore, the absorptance $\alpha = 1 - R$. The contribution of nanotubes was determined by comparing the complete sample reflectivity with the reflectivity of the stainless steel substrate alone, as presented in the Supplementary Information and Figure S1.

Emittance measurements were carried out by the Emissometer detector connected to the RD1 voltmeter reading display (Devices & Services Company, Dallas, TX, USA).

### 2.5. Thickness Measurement

The thickness of the coatings was measured by Focused Ion Beam Scanning Electron Microscopy (FIB-SEM, Model Helios Nanolab 460F1 Lite, Thermo Fisher Scientific, Hillsboro, OR, USA).

### 2.6. SEM Measurements

SEM images were obtained using a UHR-SEM (Ultra High-Resolution Scanning Electron Microscope) Magellan 400 L produced by FEI (Field Emission Instruments, Noord-Brabant, The Netherlands).

### 2.7. Adhesion Tests

Adhesion tests were performed by using the Cross-Cut Tester 1 mm according to standards ASTM D3359 [30] and ISO 2409. [31] In this test, a lattice pattern was cut into the coating, penetrating through the substrate. A tape was then placed on the cut pattern and peeled. The coating area was observed and the adhesion was rated, following a standard scale.

## 3. Results and Discussion

In our recent studies, CNT coatings were deposited on aluminum and Inconel substrates from solvent-based dispersions [27,28]. Since pipes for solar selective absorbers are usually manufactured from SS, we studied the efficiency of CNT coatings on this substrate as solar selective absorbers.

### 3.1. Coatings with Various Types of CNTs

Scheme 1 illustrates the fabrication of the CNT coatings on SS substrate. The liquid coating formulation was sprayed onto the substrate, followed by a heat treatment, to convert the pre-ceramic polymer into a ceramic matrix. Preliminary tests were performed with several types of multiwalled CNTs (MWCNTs). The most stable and low viscosity formulations were obtained with a 0.6% water-based dispersion of Cheap Tubes CNTs ($\alpha$ = 97.3%, $\varepsilon$ = 0.79) and with a 0.38% solvent-based

dispersion of Nanocyl CNTs ($\alpha$ = 94.8%, $\varepsilon$ = 0.80). These coatings have high adhesion to stainless steel substrates (95%) due to the presence of the binder.

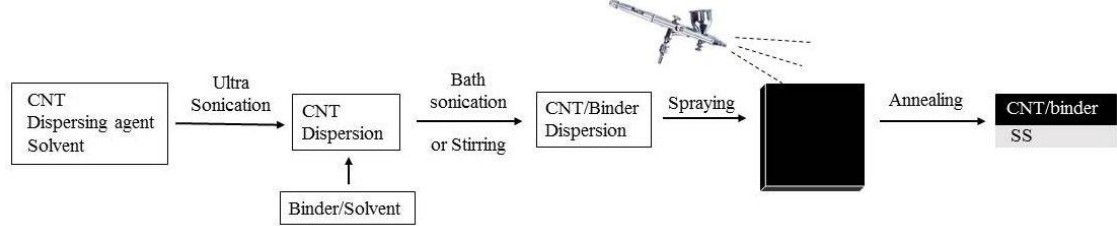

**Scheme 1.** Fabrication of carbon nanotube (CNT) coating on a stainless steel substrate.

The final absorptance of the coating depends on the CNT concentration in the dispersion, as well as on the ability of the CNTs to form a high-quality dispersion without agglomeration or rapid sedimentation. It was found that the water-based formulation must contain a higher CNT concentration to achieve the same absorptance that is obtained for a solvent-based CNT dispersion. Nevertheless, the Cheap Tubes type of CNTs was better dispersed in water, and the obtained formulations were more stable as compared with solvent-based ones.

A coating with good spectrally selective properties is characterized by high absorptance, low emittance, good adhesion to the underlying metal substrate, and high thermal stability for an extended duration. In general, all formulations and types of CNTs resulted in high absorptance, very good adhesion, and high thermal stability of the coatings. However, the disadvantage of these coatings was their high emittance (0.80).

Figure 1 shows SEM images of the coatings prepared with various CNTs dispersed in water and solvents. The diameter of the CNTs was in the range of 15–100 nm and they were densely packed and well embedded within the polymeric matrix.

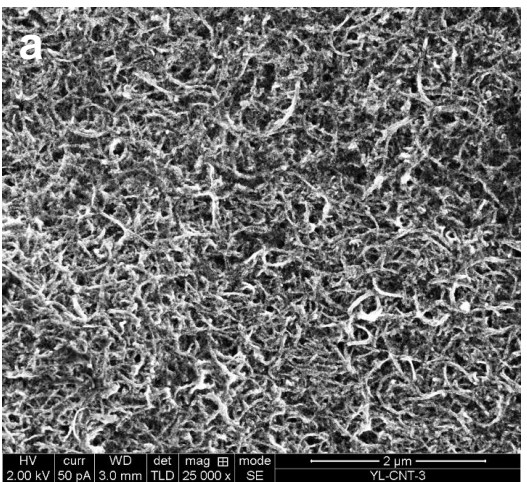
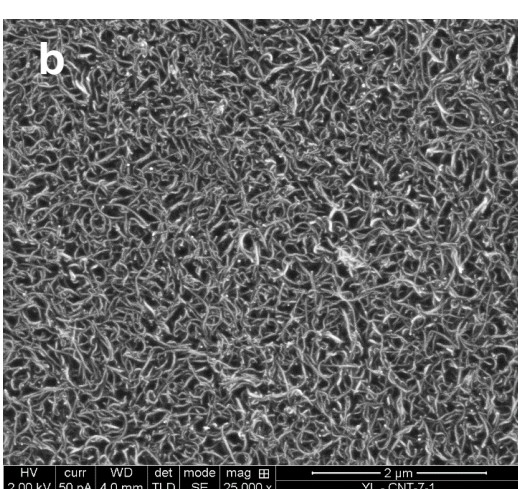

**Figure 1.** SEM images of coatings fabricated with an aqueous dispersion of Cheap Tubes CNTs (**a**), and a solvent-based dispersion of Nanocyl CNTs (**b**).

### 3.2. Coatings with a Water-Based Dispersion of CNTs

High thermal stability and good adhesion of the coatings require a binder in the formulation such as silicon resin Silres REN-100 for solvent-based dispersion and a mixture of boehmite (AlOOH): polysiloxane (MTMS) for water-based dispersion. However, it was found that the presence of a binder greatly increases the emittance of the coating. This effect was studied for various CNT concentrations, as shown in Figure 2 for coatings with and without binders.

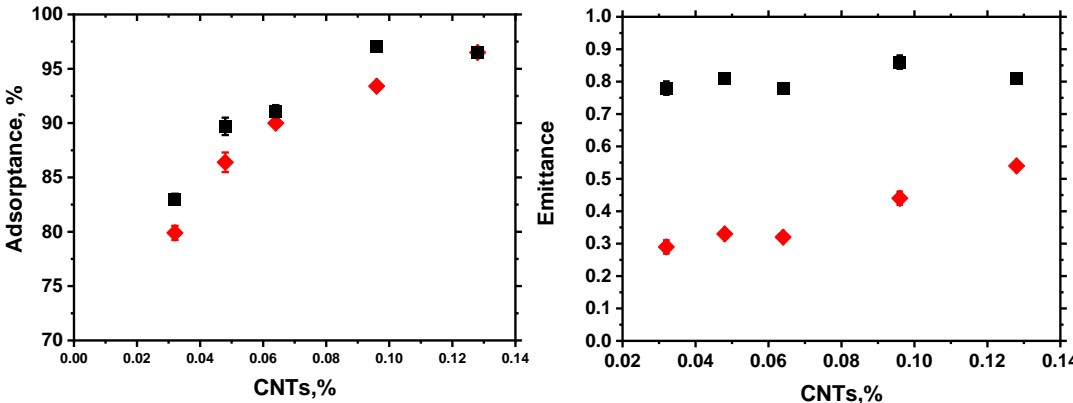

**Figure 2.** The absorptance and emittance of coatings made of various CNT concentrations, prepared with and without binder composed of boehmite (AlOOH) and polysiloxane (MTMS) with a ratio of 2:1. The binder/CNT weight ratio was 8.4:1 (◆—without binders, ■—with binders).

It can be seen that decreasing the CNT concentration led to a decrease of emittance and absorptance. Since the volumes of the sprayed formulation were constant, the above trend in absorptance and emittance was probably due to reducing the film thickness. Interestingly, there was no significant change in the emittance at various concentrations of CNTs with binders. This means that at low CNT concentrations the binder is responsible for most of the emittance. Yet, while all the coatings containing binders had good adhesion, formulations without binders had poor adhesion. Therefore, the presence of a binder is, on one hand, essential for good adhesion and, on the other hand, increases the emittance significantly. All our attempts to find a ceramic binder with significantly low emittance were unsuccessful.

Table 1 shows the absorptance and emittance of a few water-based formulations with various binders. A constant concentration of Cheap Tubes CNTs of 0.032% was used in all formulations. This concentration gave the highest difference in emissivity between formulations with and without a binder (Figure 2).

**Table 1.** Effect of binder type for water-based formulations containing 0.032% CNTs on the selective properties of the coatings. Binder/CNT ratio was 8.4:1.

| Binder | Absorptance ($\alpha$) | Emittance ($\varepsilon$) |
| --- | --- | --- |
| Without Binders | 80.7 ± 0.1 | 0.27 ± 0.02 |
| AlOOH | 86.4 ± 0.3 | 0.64 ± 0.03 |
| MTMS | 81.6 ± 0.5 | 0.66 ± 0.03 |
| AlOOH and MTMS (2:1) | 87.0 ± 0.1 | 0.78 ± 0.04 |

The results show that while the coating without the binder had an emittance of only 0.27, the addition of any of the binder components, i.e., AlOOH or MTMS, increased the emittance to 0.64–0.78. Interestingly, the presence of AlOOH [27], enhanced the absorptance. Since the presence of a binder is necessary for good adhesion to the substrate, we evaluated the effect of binder/CNT weight ratio on the properties of the coatings. These experiments were carried out with formulations containing 0.064% of CNTs (Table 2).

**Table 2.** Effect of binder/CNT weight ratio on the properties of the coatings with 0.064% CNTs. The properties of the coating with 0.128% CNTs are presented for comparison.

| CNTs, % | Binder/CNT Ratio | $\alpha$, % | $\varepsilon$ | Adhesion |
|---------|------------------|-------------|---------------|----------|
| 0.064 | 8.4:1 | 94.9 ± 0.05 | 0.60 | 65%–85% |
| | 4:1 | 93.7 ± 0.02 | 0.43 | |
| | 2:1 | 92.8 ± 0.05 | 0.38 | <35% |
| | 1:1 | 92.1 ± 0.07 | 0.39 | |
| | 0:1 | 92.3 ± 0.06 | 0.35 | |
| 0.128 | 8.4:1 | 96.0 ± 0.03 | 0.80 | >95% |

The results show that decreasing the binder/CNT ratio decreased significantly the emittance while affecting only slightly the absorptance. This decrease is presumably attributed to the thinning of the coatings. At the same time, the adhesion was negatively affected by the decrease in the binder/CNT ratio. Specifically, the adhesion with the highest binder concentration (i.e., 8.4:1 binder/CNT ratio and 0.128% CNTs) gave an adhesion of 95%, while the adhesion decreased to ca. 65%–85% at the CNT concentration 0.064% and the same binder/CNT ratio of 8.4:1. At lower binder/CNT ratios, the adhesion was poor, i.e., only 35%.

Thickness measurements carried out by FIB-SEM showed that a binder/CNT ratio of 8.4:1 gave a coating thickness of 1.1–1.2 µm, twice than that obtained without the binders (0.46–0.52 µm).

It should be noted that a twofold increase in the CNT concentration (from 0.064 to 0.128) at the same binder/CNT ratio increased slightly the absorptance (due to thickening of the coating); however, the emittance increased substantially from 0.6 to 0.8. The selectivity of the coatings ($\alpha/\varepsilon$) is presented in Figure 3. As seen, the selectivity decreased with increasing the binder/CNT ratio, even though the total absorptance slightly increased.

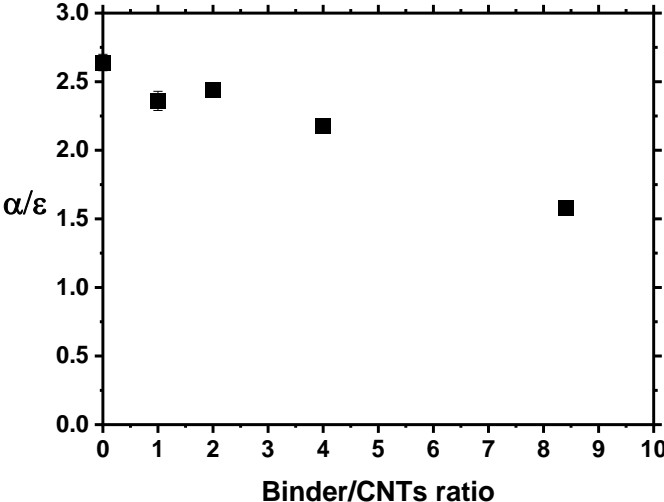

**Figure 3.** The selectivity ($\alpha/\varepsilon$) as a function of binder/CNT ratio.

The UV–vis–NIR absorptance spectra of CNTs coatings with and without a binder are shown in Figure 4. The broad absorptance band in the solar spectrum with a minimum at 1100 nm was observed for the CNTs coating without a binder, whereas introducing the binder caused an increase of the absorptance at wavelengths range between 550 and 1550 nm This explained the increase in emittance with the addition of the binder.

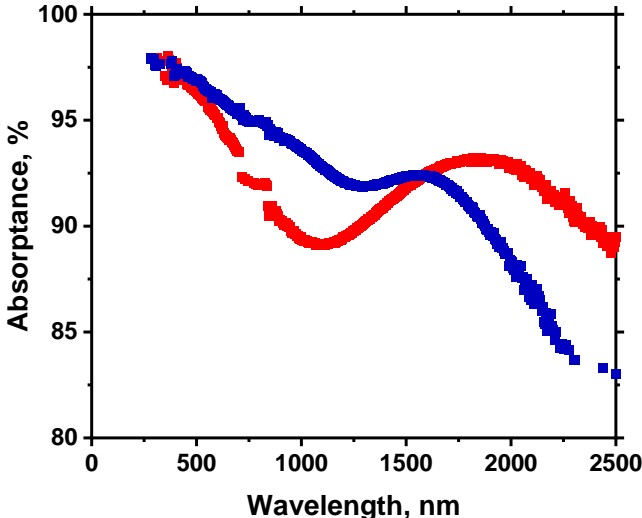

**Figure 4.** Absorptance spectra of CNT coatings without binders (red line) and with a binder/CNT ratio of 8.4:1 (blue line).

As shown in Table 1, the absorptance increased due to the presence of AlOOH in the binder's mixture. Additionally, we prepared a two-layer coating, where the first layer was made of 0.064% CNTs dispersion without the binder. The second layer was deposited by spraying on top of the first layer, a mixture of binders (AlOOH:MTMS with ratio 2:1) at various binder/CNT ratios, which was thermally treated as before (Figure 5).

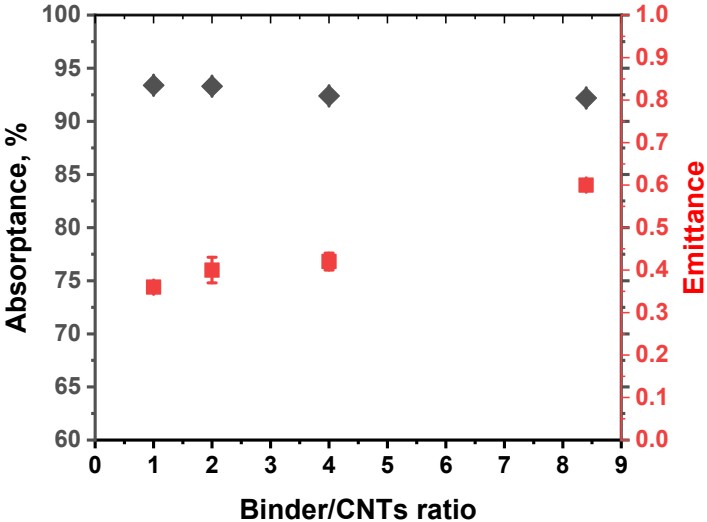

**Figure 5.** Absorptance and emittance as a function of binder/CNT ratio (calculated from the content in each layer) for a two-layer coating. CNT concentration in the formulation was 0.064%. (◆—Absorptance, ■—Emittance).

It can be seen that as the binder/CNT ratio decreased, the emittance decreased, as well, without affecting the absorptance. Yet, the binder/CNT ratio played also a major role in the adhesion to the substrate. Only coatings with a high binder/CNT ratio (8.4:1) had good adhesion (more than 95%) to the SS substrate. Hence, the binder/CNT ratio was a compromise between good adhesion and low emittance. It can be concluded that the use of water-based dispersions with a low concentration of CNTs (0.064%) makes it possible to obtain coatings with a relatively high absorptance of 92.8. Under these conditions, it was impossible to achieve both good adhesion and emittance lower than 0.59.

### 3.3. Coatings with Solvent-Based Dispersion of CNT

Since the use of formulations based on an aqueous dispersion of CNTs did not result in good selectivity, coatings based on dispersions of CNTs in an organic solvent were evaluated.

First, the effect of CNT concentration on the absorptance and emittance of coatings obtained by a solvent-based dispersion of Nanocyl CNTs without a binder was studied. As presented in Figure 6, the absorptance increased with CNT concentration and leveled off at 0.25% CNTs, where it reached 96.6%. The emittance also increased and leveled off at 0.25% and reached 0.74. These coatings adhered very poorly to the substrate and the addition of a binder was crucial.

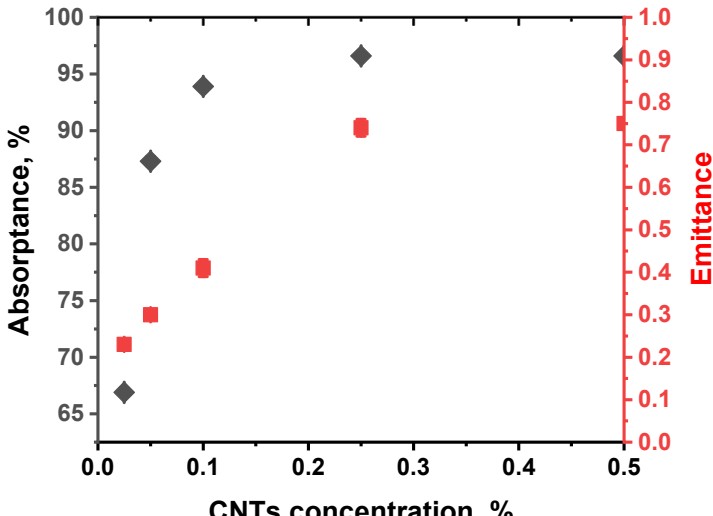

**Figure 6.** Absorptance and emittance of coatings as a function of Nanocyl CNT concentration in an N,N'-dimethylformamide (DMF)-based formulation without a binder. (◆—Absorptance, ■—Emittance).

Figure 7 shows the effect of the binder/CNT ratio on the absorptance and emittance for 0.05% and 0.1% CNTs. The binder used was a pre-ceramic polymer, i.e., Silres Ren 100. Interestingly, this binder did not affect noticeably the absorptance and emittance, as compared to the binder-free formulations, for all binder/CNT ratios. However, the binder concentration significantly affected the adhesion of the coating to the SS substrate. For 0.05% CNTs at binder/CNT ratios 1.5:1 and lower, the adhesion drastically decreased and dropped to value <35% at binder/CNT ratios 0.75:1 (Table 3). At the same time, for 0.1% CNTs, adhesion began to decrease already at a binder/CNT ratio of 2:1. Therefore, for good adhesion of this coating, the binder/CNT ratio should not be less than 3:1.

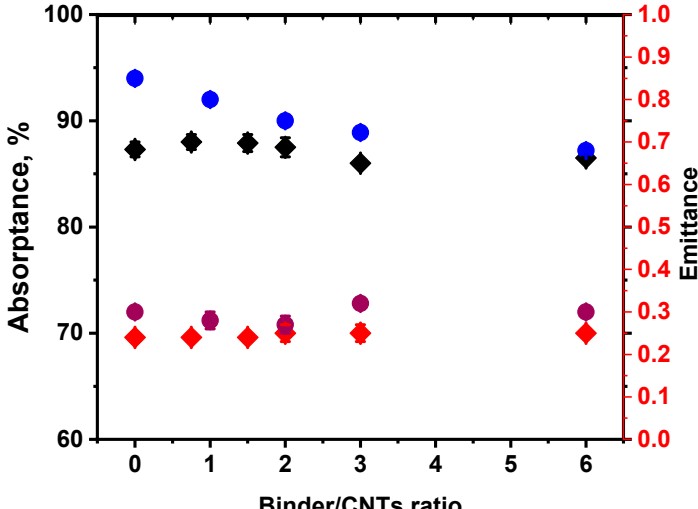

**Figure 7.** Effect of Silres Ren 100/CNT ratio on selective properties of Nanocyl's CNT coatings. ◆—absorptance and ◆—emittance for 0.05% CNTs; ●—absorptance and ●—emittance for 0.1% CNTs. The optimal coatings (with good adhesion) are marked for both 0.05% (green) and 0.1% (yellow) CNT concentrations.

**Table 3.** Effect of binder/CNT ratio on adhesion properties of CNT coatings.

| Adhesion | | Binder/CNT Ratio |
|---|---|---|
| **0.1% CNTs** | **0.05% CNTs** | |
| >95% | >95% | 6:01 |
| >95% | >95% | 3:01 |
| 65% | >95% | 2:01 |
| - | 65–85% | 1.5:1 |
| 35–45% | - | 1:01 |
| <35% | <35% | 0.75:1 |
| <35% | <35% | 0:01 |

## 4. Conclusions

Novel solar one-step fabricated coatings based on aqueous and solvent dispersions of CNTs are promising for manufacturing solar thermal collectors. A series of coating formulations containing different CNT concentrations and binder/CNT ratios were evaluated for their absorptance, emittance, and adhesion. We found that the best performance of the coatings requires adjusting the concentration of the CNTs and their ratio to the binder to obtain the highest absorptance with excellent adhesion. Specifically, high absorptance is obtained at high CNT concentration, while good adhesion requires a minimum ratio between the binder/CNT; however, increasing the binder concentration increases the emissivity. The best coatings have an absorptance of ca. 90% with an emittance of ca. 0.3 and excellent adhesion to stainless steel. Future work will focus on searching for better, lower emittance binders.

**Supplementary Materials:** The following are available online at http://www.mdpi.com/2079-6412/10/11/1101/s1, Figure S1: Reflectance spectra of a CNT coating without binders on an SS substrate (black line), and the reflectance spectra of the SS substrate alone (red line).

**Author Contributions:** Conceptualization, D.M. and S.M.; methodology, Y.V.; validation, Y.V.; formal analysis, Y.V. investigation, Y.V. and J.J.; writing—original draft preparation, Y.V.; writing—review and editing, Y.V., D.M. and S.M.; supervision, D.M. and S.M.; project administration, D.M. and S.M. All authors have read and agreed to the published version of the manuscript.

**Funding:** This research was funded by Israeli Ministry of Infrastructure and Energy, grant number 217-11-043.

**Acknowledgments:** The Hebrew University center for Nanocharacterization is warmly acknowleged.

**Conflicts of Interest:** The authors declare no conflict of interest.

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
