# Peer review of "CNT-Based Solar Thermal Coatings: Absorptance vs. Emittance"

_coatings, doi:10.3390/coatings10111101_

Round 1

Reviewer 1 Report

In the present work by Vinetsky et al., the authors investigated CNT-based coatings. A series of coating formulations containing different CNT concentrations and binder/CNTs ratios were evaluated for their absorptance, emittance, and adhesion. I do not recommend this paper for publication in the present form.

-The novelty of this work is unclear and not emphasized in the Introduction section.

-Most of the references cited here are old, which suggests that investigated topic is out of current interest.

-Line 121. It is mentioned, that in previous work the authors fabricated CNT coatings on aluminum substrate. What is the novelty of current work? Just using stainless steel instead of Al?

-Lines 24-25: the goal for 2020 declared in 2009 is mentioned. It is already 2020, more actual information should be provided.

Author Response

Attached is our reply to the reviewer

Reviewer 2 Report

The authors report on the realization and characterization of absorbing films based on carbon nanotubes. The topic is important and the paper overall good. 

I have anyway a couple of comments:

1) along the manuscript there are several typing errors. For example the absorptance is often misspelled, even in the figures labels.

2) the authors determine the absorptance as 1-R because the used substrate is not transparent. Anyway I find that the relevant quantity to investigate is the optical behavior of the nanotubes film. It is true that overall T=0, but this reasonably mainly comes from the substrate absorption. The contribution of nanutubes could be determined by using twin films on transparent substrates, or comparing the complete sample reflectivity with the one of the simple substrate. 

Author Response

Attached is our reply to the reviewer.

Round 2

Reviewer 1 Report

accept

Reviewer 2 Report

The authors revised the manuscript fully addressing my previous comments. I thus suggest the manuscript acceptance in the present form